# Homotopy Phases of FQHE with Long-Range Quantum Entanglement in Monolayer and Bilayer Hall Systems

**DOI:** 10.3390/nano10071286

**Published:** 2020-06-30

**Authors:** Janusz Edward Jacak

**Affiliations:** Department of Quantum Technologies, Faculty of Fundamental Problems of Technology, Wrocław University of Science and Technology, Wyb. Wyspiańskiego 27, 50-370 Wrocław, Poland; janusz.jacak@pwr.edu.pl

**Keywords:** correlated Hall homotopy phases, long-range quantum entanglement, bilayer Hall systems, indirect excitons, superfluidity of excitons, reentrant IQHE, FQHE hierarchy in Hall systems, control over homotopy phases

## Abstract

Correlated phases in Hall systems have topological character. Multilayer configurations of planar electron systems create the opportunity to change topological phases on demand using macroscopic factors, such as vertical voltage. We present an analysis of such phenomena in close relation to recent experiments with multilayer Hall setups including GaAs and graphene multi-layers. The consequences of the blocking or not of the inter-layer electron tunneling in stacked Hall configurations are analyzed and presented in detail. Multilayer Hall systems are thus tunable topological composite nanomaterials, in the case of graphene-stacked systems by both intra- and inter-layer voltage.

## 1. Introduction

Correlations in multiparticle systems are induced by interparticle interaction but are governed by topology and geometry. They occur in any dimensional space, but especially are observed in large number in two-dimensional spaces. The reason for this is related to the exceptionally rich topology of planar systems of identical quantum particles, which can be expressed with mathematical rigor in terms of homotopy groups [1,2,3]. In particular, the fundamental homotopy group $\pi_{1}$ of configuration spaces of multiparticle systems (known as the braid group) finds an application in the enumeration of various types of correlations in 2D electron systems exposed to magnetic field or Berry field, the latter in topological Chern insulators [4]. Classification of various correlation types in 2D interacting electron systems [5,6,7] supplements the classification of 2D crystalline structure of Bloch space of topological insulators [8,9,10,11,12]. It is commonly approved that topological degrees of freedom of multiparticle correlated systems may serve as the carriers of quantum information (QI) resistant to the local decoherence and prospective for large-scale quantum information processing (QIP). The latter is limited only to a few qubits in non-topological realizations due to so-called ’three order limit’ of local quantum mechanics, which expresses the bound of the ratio of control time-scale to the decoherence time for local quantum qubits and gates [13,14], which does not fulfill the ’six order condition’ by DiVincenzo [15] required for implementation of the quantum error correction. Topological carriers of QI are beyond this limit and are thus regarded as the most prospective for future realization of QIP. The paradigm of non-Abelian anyons has been earlier proposed [16] but its implementation is far from realization. Some attempts including Majorana fermions are still not mature [17,18].

Alternative approaches to topological QIP would offer strongly correlated states of 2D electrons in magnetic or Barry field being topologically protected against local disorder. Control over these correlations is thus of a primary importance and we aim to discuss its present status and propose some new concept of QIP organization using such topological qubits and gates. This field is naturally related with quantum Hall experiments that has recently flourished [19,20,21,22] taking advantage of new materials such as graphene multi-layers or Chern topological insulators and of great progress in manufacturing and measurement techniques.

The robustness of the correlated states of 2D electrons in Hall configuration against disorder or other decoherence perturbations is linked with a different character of the corresponding multiparticle homotopy phases in distinction to the conventional phases with a broken symmetry and a local order parameter. Famous examples of strongly correlated quantum Hall states are the integer quantum Hall states (IQH states corresponding to the integer quantum Hall effect (IQHE) in 2D electron systems exposed to the perpendicular magnetic field at integer fillings of the Landau level electron structure, discovered experimentally by Klitzing in 1980 [23] and awarded by the Nobel prize 1985) and the fractional quantum Hall states (related to the fractional quantum Hall effect (FQHE) at fractional fillings of the Landau level electron structure of 2D electrons in the magnetic field, discovered experimentally by Tsui and Stormer in 1982 [24] and explained by Laughlin in 1983 [25], also awarded by the Nobel prize 1998). The quantum states corresponding to IQHE and FQHE are not assigned by any local order parameter and any broken symmetry. Exceptions are various model anisotropic quantum Hall states that break the rotational symmetry of the system [26,27,28,29,30]. The homotopy phases for IQHE and FQHE are defined by the cyclotron commensurability of 2D electrons with the Wigner uniform distribution of these electrons on the plane [6], in a similar manner as the fractal hierarchy of Landau levels (LLs) in the periodic 2D crystal is illustrated by the Azbel–Hofstadter butterfly [31,32]. However, in distinction of the latter single-particle problem, the hierarchy of FQHE is induced by the Coulomb repulsion of electrons as the Wigner crystal is organized by the electron interaction. For the original Abel–Hofstadter butterfly the external single-particle periodic potential has been imposed on an electron and not of the Wigner crystal of electrons themselves. This makes a great difference and the cyclotron commensurability of the 2D electrons with the Wigner crystal of the same electrons is an essential multiparticle effect concerning all electrons simultaneously in a nonlocal character in contrast to single-electron problem solved by Azbel and Hofstadter [31,32]. The binary Coulomb interaction of electrons organizes via the cyclotron commensurability with Wigner lattice the correlation between simultaneously all electrons on the plane, which is in contrast to a local binary scattering of interacting electron pairs and resulting in local coupling such as, e.g., of Cooper pairs for gauge symmetry broken states of superconductors. The phases with local order parameters allow the description in terms of Landau-type quasiparticles dressed with the interaction. Quasiparticles are effective single-particle objects defined by the poles of the retarded Green function and the role of the order parameter plays the mass operator [33,34]. In correlated Hall systems none quasiparticles can be defined and the phases are characterized in the homotopy manner by braids linking the nearest or next-nearest neighbor electrons in the Wigner lattice [5,6]. This is completely different picture of the quantum multiparticle state organization which is known as the topological or homotopy state [35].

The deep fundamental difference between the homotopy phases and the conventional locally ordered phases is linked with the quantum entanglement of electrons which for the topological homotopy phase is long-range whereas for the conventional phases with broken symmetry and order parameter of is local-range only [35]. The long-range quantum entanglement concerns all electrons in the system, i.e., the multiparticle wave function is nonseparable in the multiparticle Hilbert space being the tensor product of single-electron Hilbert spaces of all electrons. The example of such a long-range entangled wave function is the Laughlin function [25], $∼\prod_{i>j}^{N}{\left(z_{i}-z_{j}\right)}^{2k+1}e^{-\sum_{i}^{N}{\left|z_{i}\right|}^{2}/4l_{B}^{2}}$ (here $z_{i}=x_{i}+iy_{i}$ denotes the 2D coordinate of *i*-th particle expressed as the complex number, $l_{B}=\sqrt{\frac{\hbar }{eB}}$ is the magnetic length at magnetic field *B*, *k* is a positive integer), which apparently cannot be separated into a tensor product of single-electron factors. Therefore, in topological homotopy phases, like at FQHE or IQHE, all electrons are quantumly entangled [35] in contrast to locally ordered conventional phases where the binary entanglement is induced by the interaction in various channels of electron scattering—like particle-particle channel of scattering and corresponding wave function describing Cooper pairs or particle-hole channel of scattering describing the normal phase of Fermi liquid [33,34].

Correlations in multi-electron systems are induced by the interaction. This refers also to such strongly correlated topological states as those corresponding to IQHE and FQHE in 2D systems at magnetic field presence. Correlations of electrons in multiparticle states are linked with quantum statistics resulting from the indistinguishability of identical particles. The statistics is imposed on the multiparticle system by the selection of the scalar unitary representation of the braid group collecting operations of exchanges of arguments of the multiparticle wave functions resolving themselves to permutations of particle indices only in 3D space or in higher dimensions. In 2D the paths of exchanges are also important, which gives an opportunity to a special kind of correlations, namely to topological correlations as those at IQHE and FQHE. Correlations are also closely linked with the quantum entanglement [35,36], which is also induced by the interaction. The quantum entanglement focused the great attention since the beginning of quantum information study [15] as the most important feature distinguishing the quantum information processing from the classical information processing. The deeply quantum resource of the entanglement is addressed in this regard to the controllable entanglement of two qubits required to construct a universal elementary gate for quantum computation [15]. From the theoretical point of view the quantum entanglement is a very simple algebraic property resulting directly from the linearity of the Hilbert space of quantum states. From this condition if follows that the Hilbert space of the multiparticle system must be of the tensor product form of single-particle Hilbert spaces of system components. Only without the interaction the multiparticle wave functions from such a tensor product Hilbert space are always separable and each particle is in a pure quantum state. Interaction causes the entanglement and for the corresponding nonseparable multiparticle wave function from the tensor product Hilbert space, the particles are in mixed states and the whole multiparticle wave function is known as quantumly entangled. It is highly interesting that the entanglement can concern all particles in the system simultaneously and not only a featured pair of particles. Such completely entangled multiparticle states occur for indistinguishable identical particles in any dimensions even without interaction. If the interaction causes strong topological correlations this entanglement of all particles still holds and is the generic property of topological phases [35]. In the case when all particles are symmetrically (or some infinite subset of all particles, in the thermodynamic limit) such a large-scale entanglement is called as long-range entanglement [35].

IQHE and FQHE have been studied since the 1980s in GaAs two-dimensional electron systems (2DES) and since ca. 2010 in graphene. The exotic nature of these phenomena was in deep recognized experimentally with still growing accuracy and was linked with specific topology of planar systems, paradoxically more rich in comparison to 3D or higher-dimensional spaces. Original discoveries of IQHE and FQHE concerned 2D electron systems in conventional semiconductor GaAs thin quantum wells [23,24]. The great new stimulus to further development of quantum 2D Hall physics was the discovery of graphene (Nobel prize 2010), which opened a new avenue for the experiment due to simple methods of manufacturing of high quality perfectly planar quantum electron systems in form of graphene sheets. Despite rapid progress in experiment and in sampling techniques the complete theoretical understanding of growing number of experimental data in Hall physics is, however, still not achieved. Though the IQHE and FQHE are both related to long-range quantum entangled states of all electrons the effective single-particle model of so-called composite fermions (CFs) [37] occurred to be useful at the early stage of the theory formulation. In must be emphasized, however, that CFs are not quasiparticles, they are not linked with any mass operator induced by the interaction—it does not exist a mass operator induced by the Coulomb interaction and dressing electrons with quanta of the magnetic flux. CFs are rather an artificial heuristic concept of somehow pinned to electrons of magnetic field flux quanta which mimics much deeper physical mechanism standing behind the FQHE. The magnetic field quanta of which are assumed to be pinned to electrons is completely fictitious and moreover, the auxiliary construction of the composites of the flux quanta with electrons is not precisely defined and only based on the idea of Aharonov–Bohm phase shift [38] when fermions associated with local magnetic fluxes exchange mutually positions. The convenient mathematical formalism of 2D Chern-Simons (Ch-S) gauge fields allows for abstract description of auxiliary field fluxes to original fermions [39,40]. It must be, however, emphasized that Ch-S field theory does not derive CFs because CH-S transformation is not canonical one and is also an artificial concept to formalize phenomenologically the quite different origin topological global effect by local by-hand manipulations with particle statistics [41,42].

The model of CFs occurred insufficient to explain fractional states in higher LLs in GaAs [43] and in bilayer GaAs Hall systems [44,45] as well as to identify so-called enigmatic states in the lowest Landau level (LLL) in GaAs (i.e., fractions $\frac{4}{11},\;\frac{5}{13},\;\frac{3}{8},\;\frac{3}{10},\ldots$ [46]). The new generation of experiments in graphene monolayer [47,48] and bilayer [21,22], also of bilayer graphene vertically gated [49] completely precluded the CF model and especially clearly manifested by themselves the topological homotopy phases as responsible for FQHE. This conveniently meets with new demonstrations of the fractional Chern insulators with typical for Hall physics hierarchy of fractions but without the magnetic field and thus without CFs for certain [50,51,52].

To display explicitly the long-range quantum entanglement in topological homotopy phases of Hall-type a different approach must be developed. The appropriate approach uses homotopy braid group theory characterizing both correlations and statistics, in particular in 2D interacting electron systems exposed to the perpendicular magnetic or Berry field. This nonlocal approach occurred helpful in the precise identification of the complete discrete hierarchy of fillings for FQHE in GaAs 2DES in the LLL including mentioned above controversial enigmatic states [6] as well as FQHE in graphene monolayer and bilayer [7,53] and in higher LLs in variety of materials [54] and recently also in explanation of astonishing FQHE states at ±1/2 and ±1/4 in monolayer graphene astonishingly substituting the Hall metal states predicted by the CF model [55].

In the present paper, we will generalize the topological approach to multilayer Hall systems with controlled regimes of the inter-layer tunneling of electrons and related changes of the collective behavior topologically conditioned on demand when electron tunneling is admitted or not by an external macroscopic factor, like by a vertical voltage. This finds the relation with the recent experiments with multilayered Hall systems illustrating the competition between the homotopy phase and locally correlated superfluid phase of inter-layer excitons [19,20,56]. Multilayer Hall systems are of particular significant as they would serve as the platform for implementation of topological gates controllable by external fields. The correlations protected by the long-range quantum entanglement is the topological feature resistant against local decoherence. The quantum states of such long-range entangled states are gaped in energy which gives the scale of their durability to the thermal chaos. Local perturbation of single electrons does not influence the homotopy phase and the corresponding quantum state in the multiparticle Hilbert space. The challenging question is to define topological qubit spanned by two distinct homotopy phases.

In the following paragraph we summarize the idea of the homotopy group approach to quantum Hall systems. Next, the application of it to the explanation of recent experiment is shortly illustrated. The following part of the paper is concentrated on multilayer Hall systems, with emphasizing of the control over topological homotopy phases as the candidates for topological qubit and QIP. Two regimes of bilayer Hall systems will be analyzed in detail and the homotopy change of the correlation is considered via control over inter-layer tunneling of carriers. The topological distinction between Hall bilayers with and without inter-layer tunneling is studied with close relation with very well developed experiment both for GaAs [44,45,56] and graphene multi-layers [19,20,21,22,49]. The multilayer graphene setups are of particular interest as the sandwich structures bilayer graphene/hBN/monolayer-graphene/hBN/bilayer graphene (hBN is the hexagonal boron nitride) offer the simultaneous control over both regimes intra- and inter-layers, which is prospective for implementation of topological QIP gate in fully accessible current technology. Some related theoretical aspects and calculation details are shifted to the Appendix A.

## 2. Main Elements of the Topological Homotopy Approach to Quantum Hall Systems

Quantum statistics of particles in many particle systems is defined by the topology of the multiparticle configuration space of indistinguishable particles moving on some manifold [2,3,57,58]. If this manifold is three-dimensional, like $R^{3}$, only bosons and fermions are possible (the same holds in higher-dimensional spaces). However, if the manifold is two-dimensional, like $R^{2}$ or finite 2D manifold (e.g., a sphere), anyons are admitted besides bosons and fermions [2,59]. Such conclusion follows from the formalism of Feynman path integral [60,61,62] applied to identical particle systems [2,3,57] where exchanges of indistinguishable particles is precisely described by the braid groups—the $\pi_{1}$ groups collecting nonhomotopic path-classes in the multiparticle configuration space [2,3,5,57]. The topological structure of these paths depends on the dimension of the manifold on which particles are placed. In 3D (and at higher dimensions) the braid groups are simple finite permutation groups, but in 2D not. In 2D the braid groups are infinite and reflect complicated patterns of trajectory entanglements impossible to disentangle without trajectory cutting. Braids are *N*-strand bundles of mutually entangled lines and in 2D it is no room to disentangle such a muddle—various patterns of braids are nonhomotopic, i.e., cannot be continuously deformed one into other without cutting. Scalar unitary representations of braid groups define quantum statistics of particles according to the mathematical structure of the path integral [2,57,60]. In other words, the one-dimensional unitary representation of the braid defines the transformation of the multiparticle wave function when arguments of this function mutually exchanges according to this particular braid [3]. For the permutation groups only two scalar unitary representations are possible, $\sigma_{i}\to e^{i0}\;\mathrm{or}\;e^{i\pi },$ corresponding to bosons and fermions, respectively ($\sigma_{i}$, i=1,…,N−1, are generators of the braid group, here the permutation group, i.e., exchanges of *i*-th particle with (i+1)-th one, at some arbitrary but fixed particle enumeration). In 2D the braid groups have infinite number of distinct scalar unitary representations, $\sigma_{i}\to e^{i\alpha }$ (where α∈[0,2π)) defining anyons [2,3,59,63]. The braid group theory is developed upon the algebraic topology [64,65,66] and linked to physics [1,2,3,59].

Fractional statistics for anyons do not exhaust, however, all physical contents related to braid groups for particles on 2D manifolds. In the magnetic field perpendicular to the plane with charge particles any braid has to be built of halve-pieces of cyclotron orbits which are of a finite size in 2D without a drift along the field direction—thus braids acquire a finite metrics. The braids describe exchanges of particle positions (arguments of the multiparticle wave function) and thus are possible to be defined only if their metrics perfectly fits to the particle separation. If the particles interact, let say electrons which repulse themselves by the Coulomb forces, the separation of them on a plane is rigidly fixed by the Wigner crystal lattice. Thus, exchanges via braids of interacting electrons are possible only in the case of the perfect commensurability of cyclotron braid metrics with the Wigner lattice of electrons. Braids define correlations and their various types display distinct patterns of the commensurability of cyclotron braids with the Wigner crystal including nearest and next-nearest neighboring electrons in this lattice and single and multi-loop braids. The resulted discrete hierarchy perfectly reproduces the FQHE hierarchy of 2D electrons [6,7]. If cyclotron braids are substituted by the Berry field orbits in 2D Chern topological insulators, the similar hierarchy of the commensurability patterns displays the fractional Chern insulator state hierarchy, with the distinction that the filling rates are counted here with respect to the crystal node occupation by electrons instead of the fractional filling of the degenerated LLs in the case of 2D Hall systems.

The cyclotron loops (or Berry field loops) are of the same finite size for electrons in the LL (or in the degenerated band of the Chern insulator) and this size is accommodated to the degenerated kinetical energy in the LL (or in the degenerated band). In particular, in the LLL the size of cyclotron loops is defined by the magnetic field flux quantum, $\frac{h}{eB}$, cf. Appendix A, and in higher LLs by $\left(2n+1\right)\frac{h}{eB}$ where *n* is the Landau index (in Chern insulator the Berry field orbit is defined by the circumference of the elementary cell in 2D crystal). This size imposes the metrics on braids and this metrics must commensurate with the interparticle spacing of repulsing electrons, as braids must realize exchanges of these electrons. When the magnetic field *B* is large enough that $\frac{h}{eB}<\frac{S}{N}$ (*S* is the 2D sample size and *N* is the total number of electrons), then simple loop-less exchanges of electrons are not admitted, as the elementary loop-less braids, $\sigma_{i}$, are too short and cannot match even the closest electrons.

However, in the braid group there are present multi-loop exchanges of electrons due to the algebraic structure of the group generated by loop-less $\sigma_{i}$—half-pieces of single-loop cyclotron orbits. These multi-loop exchanges are $\sigma_{i}^{2k+1}$, where *k* is an integer equaled to the number of additional loops in such a multi-loop braid. It is easy to see that braids $\sigma_{i}^{2k+1}$ with additional *k* loops define also the exchange of *i*-th electron with (i+1)-th one. Cyclotron braids are half-pieces of cyclotron orbits, thus 2k+1 is the number of loops in multi-loop cyclotron orbit corresponding to the braid $\sigma^{2k_{1}}$, as illustrated in Figure 1A–C (for details cf. Ref. [5,6]). (Note, that braids $\sigma_{i}^{2k}$ are not exchanges of particles as leaves particles on their initial positions, despite in 2D $\sigma_{i}^{2k}\neq e$, *e*—the neutral element in the group, in contrast to 3D where $\sigma_{i}^{2k}=e$.) It has been proved by application of the Bohr-Sommerfeld rule that the size of the multi-loop cyclotron orbit (with (2k+1) loops) is larger than the size of the single-loop cyclotron orbit and equals to $\frac{(2k+1)h}{eB}$ [6] (cf. Appendix A). In other words, the effective flux quantum for multi-loop cyclotron orbit is $\frac{(2k+1)h}{e}$ in the LLL, i.e., multi-loop cyclotron orbits and half of them—the multi-loop braids have (2k+1)-times *larger* metrics than single-loop ones. If this larger sized braids fit to the interparticle spacing, $\frac{S}{N}$, then the braid structure can be again defined and the multiparticle correlations and quantum statistics can be established, but with new braid group generators, $\sigma_{i}^{2k+1}$, instead of $\sigma_{i}$ (i=1,…,N−1). This new braid group generated by $\sigma_{i}^{2k+1}$ is the subgroup of the original one (as the new multi-loop generators are express by the original generators $\sigma_{i}$) and we call it as the cyclotron braid subgroup [5,6].

Hence, according to the Feynman path integral quantization scheme [2,3,6,57], the statistics in 2D multi-electron quantum system at large magnetic field will be defined by the scalar unitary representation of the appropriate cyclotron braid subgroup and not by an unitary representation of the initial full braid group. As usually for multi-cyclic subgroups (defined by the generators) the scalar unitary representations of cyclotron braid subgroup can be the projective representations of the full braid group, i.e., $\sigma_{i}^{2k+1}\to e^{i(2k+1)\alpha }$, when for the covering group with generators $\sigma_{i}$ the unitary representation was $\sigma_{i}\to e^{i\alpha }$. In particular, if α=π, as for original fermionic electrons, we get the cyclotron group representation, $\sigma_{i}^{2k+1}\to e^{i(2k+1)\pi }$. This means that when electrons exchange their positions on the plane at such strong magnetic field that the single-loop cyclotron orbits are too short, the phase shift of the multiparticle wave function is (2k+1)π, but only at such fields, when the commensurability $\frac{(2k+1)h}{eB}=\frac{S}{N}$ holds. The multiparticle wave function has to transform according to the scalar unitary representation of the braid group element if coordinates of this wave function exchange according to this particular braid [3] and the unitary representation of the cyclotron braid subgroup reproduces the Laughlin function phase shift [25], without the heuristic invoking to the Aharonov–Bohm phase as in the CF model.

The magnetic fields for which single-loop orbits in the LLL $\frac{h}{eB}$ are shorter than 2D electron separation $\frac{S}{N}$ and (2k+1)-loop orbits perfectly fit to this separation, correspond to fractional filling rates of the LLL, $\nu =\frac{N}{N_{0}\left(B\right)}=\frac{1}{2k+1}$, where the degeneracy of LLs, $N_{0}\left(B\right)=\frac{BSe}{h}$ (note that the degeneracy of LLs is the single-particle notion not connected with the multiparticle system and interaction of electrons, cf. Appendix A). We thus see [6] that such homotopy analysis explains the FQHE and IQHE—they correspond to strongly correlated states of *N* electrons on the 2D plane with the surface *S* for k=1,2,… and k=0, respectively, where *k* is the number of loops in elementary braids—cyclotron braid subgroup generators (or 2k+1 is the number of loops in the multi-loop cyclotron orbit). Moreover, one can state that the correlations in 2D system of interacting electrons in the presence of the perpendicular magnetic field are one-to-one defined by the cyclotron braid subgroups adjusted by the cyclotron commensurability condition to electron distribution, i.e., when the metrics of braids imposed on them by the magnetic field flux quantum perfectly fits to particle separation kept rigidly fixed by the electron repulsion. Including the nesting of braids to next-nearest electrons in the Wigner lattice and admitting the nesting separately of each loop of the multi-loop orbit, the full hierarchy of FQHE can be obtained in the full consistence with experimental observations [6]. One can thus classify the correlations in interacting electron planar systems in topological homotopy terms as shown in Table 1.

FQHE is observed in the LLL at the sufficiently strong magnetic field *B* such that the ordinary single-loop cyclotron orbit size defined as, $\frac{h}{eB}$ ($\frac{h}{e}$ denotes the universal quantum of the magnetic field flux) is smaller in comparison to $\frac{S}{N}$—the surface per single particle in the system (*S* is the planar sample surface, *N* is the number of electrons). This condition has thus the form,
(1)$$\frac{h}{eB}<\frac{S}{N}.$$
By $B_{0}$ we denote the field at which
(2)$$\frac{h}{eB_{0}}=\frac{S}{N}.$$
It is easy to notice that at $B_{0}$ one deals with the LL filling ratio, $\nu =\frac{N}{N_{0}\left(B\right)}=1$, which is clear due to Equation (Equation 2) and the definition of the degeneracy of LLs,
(3)$$N_{0}=\frac{BSe}{h}.$$
We see thus that at $B_{0}$ the IQHE realizes.

However, for $B_{1/3}=3B_{0}$ (at *S* and *N* kept constant) Equation (Equation 2) is not fulfilled any more. The inequality (Equation 1) for $B_{1/3}$ can be changed into the equality via the 3-fold increase of the quantum of magnetic field, i.e., by the substitution $\frac{h}{e}\to \frac{3h}{e}$. This enhancement of the effective quantum of magnetic field flux holds for multi-loop trajectories and can be formally proved by application of the Bohr-Sommerfeld rule [6]. Within this proof it has been shown that the effective quantum of the magnetic flux grows (2k+1) times for 2D cyclotron (2k+1)-loop orbits (where k=0,1,2…), i.e., $\Phi_{k}=\frac{(2k+1)h}{e}$—cf. Appendix A. For k=1 we get $\Phi_{1}=\frac{3h}{e}$. At the magnetic field presence the braids which describe exchanges of identical indistinguishable particles (in the classical multiparticle configuration space) are half-pieces of cyclotron orbits and therefore are of finite size like planar cyclotron orbits. The smaller cyclotron orbits have the size defined by the magnetic field flux quantum divided by the magnetic field, i.e., $\frac{\Phi_{k}}{B}=\frac{(2k+1)h}{eB}$ for single-loop orbits at k=0 and multi-loop orbits at k=1,2,…, respectively. On the other hand, any braid must start at some particle and finish at another. It means that braids (and cyclotron orbits) must be commensurate with particle distribution on the plane. The repulsing charged particles (like electrons) deposited a uniform opposite sigh planar jellium, are homogeneously distributed on the on the plane and form at T=0 K a triangle Wigner lattice. In this lattice separations between particles are rigidly fixed. Hence, the size of braids must precisely fit to this separation unless braids cannot be defined. At $B_{1/3}$ the three-loop cyclotron orbits (and related braids being the half of the cyclotron orbit) with the size $\frac{3h}{eB_{1/3}}$ perfectly fit to the particle separation $\frac{S}{N}$, i.e.,
(4)$$\frac{S}{N}=\frac{3h}{e3B_{0}},$$
though single-loop orbits are too small. For $B_{1/3}$ the LL filling rates equals to,
(5)$$\nu =\frac{N}{N_{0}}=\frac{Nh}{3B_{0}Se}=\frac{1}{3}$$
which is the most pronounced filling rate for FQHE. It is easy to notice that for next *k* and fields $B_{k}=\left(2k+1\right)B_{0}$ we get $\nu =\frac{1}{2k+1}$ and the commensurability of (2k+1)-loop cyclotron orbits with particle separation still given by $\frac{S}{N}$ which agrees with the main series of FQHE states described by the famous Laughlin wave functions of (2k+1) rank [25].

To obtain the whole hierarchy of FQHE one can generalize the above scheme of the cyclotron braid commensurability vie the inclusion of the fitting of braids with also next-nearest neighbor electrons in the Wigner lattice. This simple generalization of the commensurability condition attains the form [6,7],
(6)$$\begin{array}{c} \frac{S}{N}=\frac{h}{eBx_{1}}\pm \frac{h}{eBx_{2}}\pm \cdots \pm \frac{h}{eBx_{q}},\;\\ \mathrm{or},\;\;\;\\ \nu ={\left(1/x_{1}+1/x_{2}+\cdots +1/x_{q}\right)}^{-1}, \end{array}$$
where q=2k+1 is the number of loops of multi-loop cyclotron orbit (*k* is the number of loops in the braid), $x_{i}\geq 1$ indicates here the rank of the next-nearest neighbors which commensurate with *i*-th loop from the multi-loop orbit (in the case $x_{i}=1$ the nearest neighbor particles take part in the commensurability of the *i*-th loop), by ± we indicate the congruent, +, or inverted, −, orientation of the subsequent loop with respect to the preceding one. In the lower equation in (Equation 6) the signs ± are incorporated to $x_{i}$.

The energy gain in the Laughlin states is due to the lowering of the Coulomb repulsion energy, $\langle \Psi |\sum_{i,j}\frac{e^{2}}{|z_{i}-z_{j}|}|\Psi \rangle$, where $|\Psi \rangle =A\prod_{i>j}{\left(z_{i}-z_{j}\right)}^{2k+1}e^{-\sum_{i}{\left|z_{i}\right|}^{2}/4l_{B}^{2}}$ is the Laughlin function of *k*-th rank (A is the normalization constant, $l_{B}$ is the magnetic length). The Jastrow monomial factors ${\left(z_{i}-z_{j}\right)}^{2k+1}$ indicate the rising with *k* separation of *i*-th and *j*-th particle density distribution, which causes reduction of the averaged Coulomb repulsion energy. It is clear that this energy reduction prefers thus the closest correlations of particles (which corresponds to correlations of all particles equally in the system), thus the best if $x_{1}=1$ in the above formulae. If the correlations do not concern the closest neighbors at all (for example for $x_{i}=2$ there are correlated only every second particles, i.e., only N/2 subsystem of all electrons), then similar energy preference causes all $x_{i}>1$ to be as minimal as possible, hence (due to increasing ordering of these parameters) one can choose $x_{1}=\cdots =x_{q-1}=x$ and $x_{q}=y\geq x$. Such an argumentation holds in GaAs where the envelope function in the multiparticle wave functions (this beyond the polynomial part) is the same as in the gas and prefers the nearest neighbor correlations. Thus, for GaAs the commensurability condition attains the simplified form,
(7)$$\frac{S}{N}=\frac{(q-1)h}{eBx}\pm \frac{h}{eBy},$$
which defines the hierarchy of FQHE in the LLL in GaAs,
(8)$$\nu =\frac{N}{N_{0}}=\frac{yx}{(q-1)y\pm x},$$
with q=2k+1 odd integer and y≥x≥1 positive integers 1,2,…. For x=1 the hierarchy (Equation 8) gives the Jain hierarchy for CFs [37]. CFs reflect thus the commensurability condition for *q*-loop orbit, but only if q−1 loops are nested to the nearest neighboring electrons (x=1). Note, however, that in graphene monolayer the envelope function is distinct than that for the gas and can prefer next-nearest neighbors at certain windows for the magnetic field [47], which does not allow the above simplification [55].

We see thus that the CF model is a pictorial effective illustration of the cyclotron multi-loop braids but only if the braid nesting concerns the nearest neighbors for 2k loops of q=2k+1-loop cyclotron orbit. The fictitious auxiliary field postulated in the CF model to be fixed to moving electrons, is an effective model of the additional cyclotron loops. Each additional cyclotron loop is modeled in the CF theory by adding to the electron of one quantum of the fictitious field. As the lowest number of additional braid loops is k=1, thus number of additional loops in the cyclotron orbit must be 2k=2, and 2 fictitious flux quanta are added to the simplest CF. It is clear that CFs are not a real quasiparticles, they are auxiliary objects as fictitious as is the magnetic field, quanta of which are pinned to CFs. Such a field does not exist in the system, and it only mimics additional braid loops. The CFs illustrate in heuristic single-particle manner the more fundamental topological multiparticle property of multi-loop braids in 2D interacting electron systems at the magnetic field presence.

The general hierarchy (Equation 8) gives all FQHE filling ratios experimentally observed in GaAs in the LLL, including so-called enigmatic states (those out of CF model), e.g., $\nu =\frac{4}{11},\frac{5}{13},\frac{3}{8},\frac{3}{10},\ldots$ [6,46]. The enigmatic states filling ratios are included to the hierarchy (Equation 8), but at x>1 (thus, they are apparently not of CF type).

### 2.1. Homotopy Approach to Fractional Hall States in Higher LLs of GaAs

The hierarchy of fractional fillings in higher LLs of GaAs 2DES does not repeat the hierarchy pattern form the LLL. In both spin up and down sub-bands of the LLL the FQHE hierarchy is identical, but for Landau index n≥1 the experimentally observed structure of FQHE is completely different than for n=0 [43,67]. It is easy to explain such an odd behavior in homotopy terms (but is not possible in the CF model). In higher LLs the kinetical energy of electrons is greater, $E_{n}=\left(2n+1\right)\hbar \omega_{0}$ (where $\omega_{0}=\frac{eB}{2m}$ and *n* is the Landau index enumerating LLs).

To larger kinetic energy in higher LLs correspond larger cyclotron orbits, $\left(2n+1\right)\frac{h}{eB}$ for single-loop ones, and $\left(2n+1\right)\frac{(2k+1)h}{eB}$ for (2k+1)-loop ones. If the orbits fit to particle separation in higher LLs, e.g., in the first spin sub-band of the first (n=1) LL, to the separation, $\frac{S}{N-2N_{0}}$ (because two spin sub-bands of the LLL are completely filled up to the degeneracy $N_{0}\left(B\right)$ of each LLL sub-band at field *B*, only $N-2N_{0}$ electrons left to fill the next sub-band with n=1), then the correlations in the form of FQHE state in this sub-band are possible when cyclotron orbits nest to interparticle spacing. Nevertheless, in higher LLs the single-loop orbits (with k=0) are sufficiently large, because n≥1, and the single-loop orbits allow nesting with nearest and even next-nearest neighbors, which was not possible in the LLL (in the LLL single-loop cyclotron orbits were always shorter than the interparticle separation, in higher LLs not). The commensurability of single-loop orbits in higher LLs with nearest and next-nearest neighbors perfectly reproduces the experimental data for FQHE in higher LLs in GaAs 2DES [54]. In particular in the first LL with n=1 in both its spin sub-bands the doublets of FQHE states at $\nu =\frac{7}{3},\;\frac{8}{3}$ and $\frac{10}{3},\;\frac{11}{3}$ result from the single-loop commensurability with nearest and next-nearest neighboring electrons but not from three-loop orbit commensurability as in the LLL for fractions with the denominator 3.

Please note that in higher LLs at the vicinity of sub-band edges (i.e., for very small filling of the LL sub-band with electrons or holes) the separation of diluted particles may be so large that again multi-loop braids (and related multi-loop cyclotron orbits) would be needed, but in the central regions of sub-bands of higher LLs, the single-loop orbits are sufficiently large to nest with the particle separation.

All these predictions agree with experiments. Moreover, the braid commensurability criterion gives at centers of sub-bands of higher LLs (for n≥1) the paired states [7], whereas in the LLL at ν=1/2 and 3/2, not paired compressible Hall metal state [68]. The full hierarchy of Hall metal states in all LLs is found by the limit to the infinity of the size of one from loops of multi-loop orbits, which perfectly reproduces experimental observations [68].

### 2.2. Explanation of ±1/2 and ±1/4 States in Monolayer Graphene in Some Magnetic Field Window

In monolayer graphene the FQHE hierarchy should repeat the hierarchy patterns as in GaAs with the same modification as for IQHE in monolayer graphene reflecting the spin-valley 4-fold splitting of LLs into sub-bands [69]. The another difference is connected, however, with the distinct single-particle LL states in monolayer graphene than in GaAs. This single-particle wave functions in monolayer graphene are eigen functions for the Landau problem but in graphene must be included the hexagonal crystal lattice and related crystal field (note that the latter is also a single-particle notion and not influences the homotopy of braid trajectories). The resulted LLs in graphene are different than those in GaAs, and the eigen energies are, $E_{n}=\hbar \omega^{\prime }\sqrt{n}$, $\omega^{\prime }=2v_{F}/l_{B}$ [69] (where $v_{F}$ is the Fermi velocity, $l_{B}=\sqrt{\frac{\hbar }{eB}}$ is the magnetic length), thus are not linear with respect to the Landau index *n* as they were in GaAs. However, the degeneracy of LLs is the same as in GaAs, $\frac{BSe}{h}$ (the universality of the LL degeneracy is commented in Appendix A). The crystal field is the local electric type field, which does not influence the homotopy classes for multiparticle correlations in magnetic field, which in graphene monolayer are the same as in GaAs. The theoretical hierarchy of FQHE is thus identical in GaAs and in graphene monolayer, which some reservation, however. This reservation is related with different envelope of the multiparticle wave functions for particular cyclotron homotopy classes, though the homotopy classes are same as in GaAs. These envelopes are invariant against particle exchanges and therefore can be different in graphene and substitute the envelope in GaAs, $e^{-\sum_{i}^{N}{\left|z_{i}\right|}^{2}/4_{B}^{2}}$, where $z_{i}=x_{i}+iy_{i}$ is the complex coordinate of *i*-th electron, i=1,…,N and $l_{B}=\sqrt{\frac{\hbar }{eB}}$ is the magnetic length. The latter in GaAs corresponds to conventional LLL states of the free electron (the hypergeometric confluent function in the LLL of an electron at symmetric gauge for *B* field). In graphene this envelope is different due to the crystal field, though the multiparticle function core sensitive to the electron interchanges is the same in graphene and GaAs, as defined by the scalar unitary representation of particular cyclotron braid subgroup and the explicit form of this subgroup generators, identical in both materials [6]. The core function has in the LLL the form of the homogeneous polynomial. The envelope can, however, change the eigen energy for particular states and does it in a different manner in graphene in comparison to GaAs. Thus, some states from the common hierarchy can be suppressed in energy in favor to another ones, changing the experimental manifestation of the same hierarchy. In experiment there are visible the most convenient energetically states at each filling rate admitted by the commensurability condition. Moreover, some filling rates, which are unstable in GaAs (have lower energy) can become stable in monolayer graphene, and conversely. As has been demonstrated in [47] the states at ν=±1/2 and ±1/4 occur in monolayer graphene as FQHE states, though both are Hall metal states in GaAs. The clue of such queer behavior reflects the situation that in general and common for both materials the states at ν=1/2,1/4 are Hall metal (compressible) states for nearest neighbor nesting, whereas they become of FQHE type (incompressible) when the nesting concerns next-nearest neighbors (cf. Table 2 and Figure 1). As is visible in Table 2, the same filling ratios can correspond to various homotopy classes and the energy chooses the stable one. The envelope function in GaAs prefers Hall metal at 1/2 and 1/4, but in monolayer graphene for some window for the magnetic field (in Landau fan diagram) [47] the distinct envelope prefers correlations including next-nearest neighbors, and in this field windows FQHE occurs at ±1/2 and ±1/4 in graphene monolayer. Please note that because the next-nearest neighbors correlations are beyond the illustration ability of the CF theory, the described above FQHE states at ±1/2 and ±1/4 in monolayer graphene are not of CF type.

### 2.3. Homotopy Phases in Bilayer Graphene

In bilayer graphene the homotopy situation is different than in monolayer Hall systems. Even if bilayer graphene is considered to be a single Hall system it is not perfectly two-dimensional. Braids may hop between two layers because electrons can tunnel across the small barrier separating the layers. In the case of the LLL when the single-loop orbit is smaller than the electron separation, the required multi-loop cyclotron orbit may have a different character in comparison to monolayer Hall systems. Loops (or its fragments) from the multi-loop orbit may reside in both layers simultaneously. Besides the multi-loop orbits entirely located in a single layer, there are possible orbits which share their loops among both layers. In particular there are possible 3-loop orbits with two loops in one layer and one loop in an opposite layer. Thus, in the commensurability condition describing nesting with electron separation in one layer will participate two loops (not three)—which gives the FQHE state at 1/2 instead of 1/3, as indeed has been observed experimentally [22]. The commensurability condition of multi-loop cyclotron orbit with electron distribution describes the division of the total flux of external magnetic field per each loop. As both layers in bilayer graphene have separated own surfaces, thus through each layer passes independent flux of the external field *B*. Therefore, the commensurability condition may concern only single layer independently of the opposite one, which agrees with the experimental observation of 1/2 FQHE state in bilayer graphene [22].

However, in bilayer graphene the single-particle Landau-type problem has an additional specificity—the LLL is eight-fold degenerated, four-fold due to spin-valley degeneracy (similar to monolayer graphene) and two-fold one, known as accidental, corresponding to the degeneracy in bilayer graphene of oscillator-type Landau states with indices n=0 and n=1 [70]. In bilayer graphene both these states, with n=0 and n=1, belong to the LLL resulting in its 8-fold degeneracy. These states are divided by half between valence and conduction bands. The accidental degeneracy causes some consequence depending on the order of fillings of these sub-bands of the LLL with electrons, when the degeneracy is lifted. If first is filling the state with n=0 then the FQHE occurs at 1/2, as explained above. However, if first is filling the state with n=1, the FQHE state 1/2 disappears and is again substituted by the 1/3 state. This is caused by the fact that at n=1, the cyclotron orbits are 2n+1=3 times larger than at n=0 and multi-loop orbits are not needed at n=1 (similarly as in higher LLs of GaAs or monolayer graphene). This dismisses the state 1/2 related to 3-loop orbit in bilayer but allows for the states 1/3 and 2/3 single-loop ones (as in first LL of GaAs or monolayer graphene) corresponding to nesting of single-loop orbit with next-nearest neighbors. These states are not of CF type (they are not multi-loop). Such a situation is illustrated by experiment with bilayer graphene supported by boron nitride substrate (hBN) [71], which apparently converted order of almost degenerated LLL sub-bands in comparison to free suspended sample as in the experiment Ref. [22], and in the bilayer graphene sample supported by hBN the FQHE states 1/3 and 2/3 are visible and not 1/2 [71] quite opposite to the suspended sample [22].

In bilayer graphene in higher LLs the situation with hoping of braid trajectory repeats and the perfect consistence with experimental observations [21] has been obtained up to eighth LL sub-band upon the homotopy analysis of loop distribution between layers [7], i.e., in all experimentally observed FQHE features in bilayer graphene.

All these phenomena with braid trajectory hoping are out of reach for CF theory and therefore this theory completely fails in bilayer graphene. Together with other argument presented above one can repeat that CFs cannot be treated as quasiparticles (occurring only in the LLL and out of fillings of enigmatic states)—CFs illustrate cyclotron braid trajectory nesting in the exclusive case of multi-loop orbits when only nearest neighbors are taken into account. Each additional loop in the multi-loop cyclotron orbit is modeled in CF theory by the flux quantum of the fictitious auxiliary magnetic field pinned to electrons according to the idea of CFs [72]. In fact, none such field exists and CFs also do not exist—flux quanta model rather additional loops of multi-loop cyclotron orbits (or cyclotron braids). The Aharonov–Bohm phase shift at exchange of CFs [38,59] is in fact given by the scalar unitary representation of cyclotron braid subgroup in fully consistence with the Laughlin function requirements [25].

## 3. Various Regimes of Bilayer Hall Systems

The presented above braid group approach to 2D electron correlations in FQHE and IQHE in interacting planar electron systems in strong perpendicular magnetic field is especially convenient to analyze in more detail the behavior of multilayer Hall systems when the hopping of braid trajectories creates new possibilities for the nesting of braids to the particle separation in distinction to the monolayer systems. It is clear that the perturbation or the blocking of the trajectory hopping would have some consequences and would change the homotopy classes admissible or not in such a situation. This is an exceptional opportunity to control over the quantum homotopy phases of multiparticle system on demand. This topology control can be realized by the blocking or not of the inter-layer trajectory hopping by the perturbation of the tunneling of electrons between layers. This can be done by various means. One can apply a vertical voltage and block tunneling of electrons one-way along the voltage orientation. If electrons cannot tunnel in both directions between two layers, then the trajectory cannot jump at all, because the braid trajectory is closed and must come back if jumps. Thus, by an application of the vertical voltage to the bilayer graphene sample one can change its homotopy class from the bilayer to monolayer one.

The relevant experiment has been performed [49]—which demonstrates that the state FQHE at 1/2 in bilayer graphene (not of CF type) is dismissed by the application of ca. 1.5 V vertical voltage, in favor to FQHE 1/3 state (of CF type as in monolayer).

In view of this result one can clarify some relatively old pioneering experiments with conventional GaAs bilayer Hall systems [7,53]. They were puzzling since 1992 and not understood upon CF model. In the experiment [7] the manufactured double well in the GaAs/GaAlAs heterostructure has been examined at various thickness of the GaAlAs barrier separating two 2DES GaAs planes.

In the experiment [53] a sophisticated two layer Hall system has been realized in a wider well in GaAs heterostructure where due to transverse in the well repulsing of electrons, the electron shift symmetrically towards well borders resulting in effective two layer Hall system with electron density transverse profile of double well shape with the medium-height barrier in-between.

In both experiments there have been observed similar FQHE states as in bilayer graphene, though without a specific for graphene spin-valley degeneracy structure of LLs. In particular, the accidental degeneracy of n=0 and n=1 states does not exist here, which means that 1/2 state cannot be here substituted by the state at 1/3, as may happen in bilayer graphene due to lifting the degeneracy of n=0 and n=1 states into oppositely ordered of the split states depending on the bilayer graphene substrate [68]. And indeed, in both experiments [7,53], the state 1/2 is always visible, unless the thickness of the barrier precludes the electron tunneling. Moreover, the other FQHE features typical for bilayer homotopy agree with experiment observations. The phase transition of FQHE hierarchy when the barrier between Hall layers is strengthened and blocks the inter-layer tunneling is consistent with the braid homotopy change from the bilayer to monolayer type.

For the case when electrons may hop between layers, like in bilayer graphene or in the case of two closely adjacent layers of 2DEG GaAs separated with a small barrier admitting the tunneling, the total system behaves as a uniform one characterized by the total filling $\nu_{T}$ which is, however, split symmetrically for $\nu_{bot}=\nu_{top}=\nu_{T}/2$ counted per layer, here the subscripts bot and top refer to bottom ant upper layer, respectively.

### 3.1. Bilayer Hall System with Tunneling of Electrons between Layers

The bilayer Hall system with admitted inter-layer tunneling of electrons constitutes an unified Hall system for which the filling rate can be counted as in a single 2D system, $\nu_{T}=\frac{N}{N_{0}}$, where *N* is the total number of electrons (summed over both layers) and $N_{0}$ is the degeneracy of 2D Landau level (LL) sub-bands, $N_{0}=\frac{BSe}{h}$, where *S* is the surface of the sample (i.e., the surface of the *single* layer), *B* is the external magnetic field. Thus, at the total filling rate, $\nu_{T}$, per each layer falls $\nu_{top}=\nu_{bot}=\nu_{T}/2$.

Fractional hierarchy of fillings for correlated states in all Hall systems is conditioned by the commensurability of cyclotron orbit size, $\frac{\Phi }{B}$ (where $\Phi =\left(2n+1\right)\frac{(2k+1)h}{e}$ where *n* is the Landau index, whereas *k* is the multiplicity of cyclotron orbit) with separation of 2D electrons rigidly fixed by the Coulomb repulsion in the form of the Wigner crystal [6,7]. The commensurability patterns determine distinct homotopy phases appropriately to the value of the effective quantum of the magnetic field flux admitted at the particular homotopy patterns [6] in the form,
(9)$$\Phi \left(n=0\right)=\left\{\begin{array}{c} \Phi_{0}=\frac{h}{e}\;\;\mathrm{in}\mathrm{simply}\mathrm{connected}\mathrm{space},\\ \Phi_{k}=\left(2k+1\right)\frac{h}{e}\;\;\mathrm{in}\mathrm{multiply}\mathrm{connected}\mathrm{space}. \end{array}\right.$$
In (Equation 9) *k* numerates the number of loops in braids linking the electrons on the plane. The separation between repulsing electrons placed on the positive uniform jellium is given by $\frac{S}{N}$ (in each single layer of the bilayer system the separation of electrons is $\frac{2S}{N}$ because each single layer is filled with N/2 electrons only).

Due to tunneling of electrons between both layers also the trajectory can hop between these layers. Thus, the loops of the multi-loop cyclotron orbit can be distributed among layers. Please note that Loops in each layer have own surface there and must participate in the commensurability separately in layers, because across each layer passes separately the same external magnetic field flux. One can consider thus q-loop cyclotron orbit with i=0,1,…,q−1,q loops escaped to the opposite layer. In such a case in the former layer stay q−i loops. The total flux of the external magnetic field *B* passing this layer must be shared among q−i loops, $BS/N^{\prime }=\left(q-i\right)\frac{h}{e}$, $N^{\prime }=N/2$, which gives $\nu_{top(bot)}=\frac{N^{\prime }}{N_{0}}=\frac{1}{q-i}$ and $\nu_{T}=\frac{N}{N_{0}}=\frac{2}{q-i}$, as the universal degeneracy of LLs is $N_{0}=\frac{BSe}{h}$.

In the simplest case of three-loop cyclotron orbit the distribution of loops among layers can be as follows, 1−2, 0−3, 2−1 or 3−0. Thus, the strongest Hall correlations occurs at $\nu_{top(bot)}=\frac{1}{2}$ or $\nu_{top(bot)}=1$ instead of $\frac{1}{3}$ (as in the monolayer for q=3 orbit) [6]. These two homotopy patterns, $\nu_{top(bot)}=\frac{1}{2}$ or $\nu_{top(bot)}=1$, give $\nu_{T}=1$ or $\nu_{T}=2$ (because layers are symmetric). The decomposition of the three-loop orbit, 3−0 (or 0−3) gives $\nu_{top(bot)}=\frac{1}{3}$ and $\nu_{T}=\frac{2}{3}$. In the case or five-loop cyclotron orbit (q=5), the possible decompositions are, 5−0, 4−1, 3−2, 2−3, 1−4 (and mirrored), which lead to $\nu_{top(bot)}=\frac{1}{5},\frac{1}{4},\frac{1}{3},\frac{1}{2},1$ resulting in $\nu_{T}=\frac{2}{5},\frac{1}{2}$ and again $\nu_{T}=\frac{2}{3},1,2$. These all filling rates are observed in experiments in GaAs bilayer [44,45] as shown in Figure 2 and in the graphene bilayer [22,71].

For more detail of the FQHE hierarchy in the bilayer graphene in the LLL and in higher LLs cf. Ref. [7].

Worth mentioning is the difference between the bilayer GaAs Hall system and bilayer graphene. In the latter case the accidental degeneracy of n=0 and n=1 occurs which is, however, no case for GaAs. Thus, for the bilayer of GaAs the FQHE at $\nu =\frac{1}{2}$ always should be observed unless the tunneling of electrons is prohibited. This FQHE state at $\nu =\frac{1}{2}$ has been indeed observed in GaAs bilayers [44,45] (cf. Figure 2). In bilayer graphene the FQHE at $nu=\frac{1}{2}$ occurs conditionally in dependence of the way in which the accidental degeneracy is lifted. If the state with n=0 is lower in energy then the FQHE at $\nu =\frac{1}{2}$ is visible, but if the state with n=1 has a lower energy, then this state disappears. Such a behavior is visible in the experiment for bilayer graphene samples deposited on different substrates (hBN substrate or suspended in the air).

The FQHE state at $\nu =\frac{1}{2}$ was observed also [44] in the sophisticated bilayer system obtained in a wider well GaAs/AlGaAs (with the width of order of several dozen nm) which if filled with electrons spontaneously results in two symmetrical layers of electrons located closely to the well edges because of the electron repulsion. In Ref. [44] such bilayer GaAs system was examined, with the well width 68 nm—the resulting effective potential profile in the well is shown in Figure 2b). The thickness of layers is here approximately $\lambda ≃2l_{B}$ and the layer separation *d* varies between 48.5 nm to 44.5 nm for the density of electrons $3.1\times 10^{11}$/cm${}^{2}$ and $1.8\times 10^{11}$/cm${}^{2}$, correspondingly. Please note that $d/l_{B}∼6$ ($l_{B}$ is taken at B=15 T) is here much larger in comparison to 2DES system separated by a thin insulator barrier [45] (cf. Table 3). This more extended bilayer system keeps the characteristic fractional correlations and the FQHE at the fraction $\nu_{T}=\frac{1}{2}$ is visible, which is caused by the tunneling of electrons across the flat barrier despite its larger width. This state $\nu =\frac{1}{2}$ disappears at only $d/l_{B}>8$ at $\lambda <2.6l_{B}$ [44]. Observation of Hall correlated states $\nu_{T}=1,2,\frac{2}{3},\frac{4}{3},\frac{4}{5}$ has been also reported for this system [44]. The tilting of the external magnetic field to the angle $>10^{o}$ quenches the state $\nu_{T}=\frac{1}{2}$ which can be interpreted as the evidence of the decisive role of the inter-layer electron tunneling partly blocked by in-plane component of the tilted field [44]. The disappearance of the state $\nu_{T}=1$ at the tilted field (at angle $31^{\circ }$) has been also noticed [44]. The state $\nu_{T}=1$ disappears for larger layer separation, d=48.5 nm, i.e., for larger density $3.1\times 10^{11}$/cm${}^{2}$, as such large barrier blocks tunneling.

### 3.2. Double-Layer Hall System with Insulating Barrier Precluding Inter-Layer Tunneling of Electrons

The second regime of adjacent Hall layers is the bilayer system with the insulating barrier which blocks inter-layer tunneling of electrons. The related experiment has been developed [19,20,56,73,74,75,76,77,78,79] and some specific collective effects have been demonstrated including, electron drag current, superfluidity of indirect excitons at complementary filling of layers, $\nu_{T}=\nu_{bot}+\nu_{top}=1$ (or other integer), competing with the IQHE reentrant at such fractional fillings of layers.

Prohibiting of the inter-layer tunneling of carriers causes that homotopy of bilayer system is that one of two monolayers, though with electrons still interacting by Coulomb forces across the barrier. At complementary filling of both electrically coupled layers this interaction across the barrier causes two effects, (1) the strengthening of the reentrant IQHE in the whole system, and (2) Bose-Einstein condensation (BEC) of electron-hole pairs across the barrier leading eventually to a superfluid state. Both these competing phenomena lead to phase transitions observed experimentally [19,20,56,73,74,77,79].

If electrons and holes are located in one layer then the quick recombination of e-h pairs prohibits creation of the exciton Bose-Einstein condensate and its hypothetical superfluidity [80,81,82,83,84]. The Bose-Einstein condensate of excitons has been intensively searched in many systems [85,86,87,88,89,90], but the quick recombination of ordinary excitons usually disrupts the condensation. The separation of electrons and holes in the opposite layers in bilayer structure enhances, however, the stability of coupled electron-hole pairs. The insulating barrier of thickness $∼l_{B}=\sqrt{\frac{\hbar }{eB}}$ (the magnetic length) is large enough to block inter-layer tunneling but still allows the formation of inter-layer excitons due to Coulomb coupling of electrons and holes across the barrier. The Bose-Einstein condensate of such inter-layer, so-called indirect excitons and their superfluidity is believed to be observed in twin GaAs Hall planes separated by GaAlAs or in twin graphene systems (both monolayer and bilayer) separated by the sufficiently thick hBN layer, at the complementary filling of layers, $\nu_{top}+\nu_{bot}=1$. The latter condition means that the number of electrons (holes) in the Landau band in the top Hall layer fits to the number of the Landau band holes (electrons) in the bottom Hall layer [19,20,56,73,74,75,76,77,78,79]. In these experiments it is demonstrated the competition of the inter-layer exciton superfluid phase versus the incompressible IQHE state of the total system. Since both layers are fractionally filled upon the complementary scheme, the eventual IQHE state must be of the reentrant type, i.e., manifesting homotopy correlations as for ν=1 but in stripe structure when electrons are shifted and concentrated on smaller area of stripes up to local density corresponding to complete filled LLL, despite nominal fractional filling of the layer (cf. Figure 3a)).

This reorganization of local filling with electrons is energetically inconvenient due to the electrostatic interaction. However, the energy gain of IQHE in stripes prevails the energy cost of the electron redistribution. Such a reentrant IQHE is typically observed in monolayer GaAs 2DES at fractional nominal filling rates up to even ν=0.8 [46], cf. Appendix A for the detailed illustration of such reentrant IQHE (e.g., Appendix A).

In a twin Hall structure the region of IQHE reentrance is increased due to the lowering of the electrostatic energy cost of striping induced by the attraction of oppositely filled stripes across the barrier (cf. Figure 3a)). This energy gain is so large that the reentrant IQHE in bilayer may arise at arbitrarily fractional filling of layers provided the complementary scheme of fillings, $\nu_{bot}+\nu_{top}=1$, allowing for mirror symmetry across the barrier between striped layers with simultaneous change sign of carriers [19,20,56,73,74,75,76,77,78,79].

This multiparticle IQHE-reentrant correlated state competes with the superfluid phase of BEC of indirect excitons occurring also at the complementary filling of both layers. Stability of particular phases depends thus on the energy competition between both phases. By application of the Metropolis Monte Carlo (MMC) method [91] it is possible to assess the energy gain due to IQHE correlation in the striped structure when stripes ale aligned in both layers in a mirrored manner what is possible at the complementary filling of layers, $\nu_{top}+\nu_{bot}=1$, as shown in Figure 3a). The contribution of the electrostatic interaction is reduced by the Coulomb attraction of stripes across the barrier. The overall energy in the IQHE-reentrant phase can be precisely evaluated (cf. Appendix A for details),

The same structure of striping is also optimal for the competing phase of superfluid BEC of indirect excitons. Inter-layer excitons need facing of electrons and holes across the barrier. Stripe structure allows such an alignment and appropriate spatial redistribution of quantum states of carriers.

For Landau gauge of the magnetic field, *B* (B=∇×A, A=(0,−Bx,0)), the wave functions of Landau particles are as follows,
(10)$$\begin{array}{c} \psi_{k,n,\lambda }\left(x,y,\sigma \right)=Ce^{iky-{\left(x+\alpha k\right)}^{2}/2}w_{n}\left(x+\alpha k\right)\delta_{\lambda ,\sigma }, \end{array}$$
where $w_{n}$ is the Hermite polynomial, α=1 for electrons and α=−1 for oppositely charged holes (holes are here considered to be unfilled states in the LL sub-band, are positively charged due to the positive jellium), λ(σ)=1,2 distinguishes between the top and bottom layers (magnetic length $l_{B}=\sqrt{\frac{\hbar }{eB}}$ is put 1 in (Equation 10), for abbreviation). From Equation (Equation 10) it is visible that electrons and holes in the opposite stripes have perfectly mirrored alignment of states if *k* is changed for −k, i.e., electron and holes are oppositely ordered along the *x* direction, which is admissible due to degeneracy of LLs. The density of charge of the negatively charged electron *k* in upper layer is located perfectly opposite the same shape density of positively charged −k state of the hole in bottom layer, or vice versa, cf. Figure 4. These charge distributions attract mutually themselves across the barrier of thickness *d* with energy $≃-\frac{e^{2}}{4\pi \varepsilon_{0}\varepsilon_{2}d}$ ($\varepsilon_{2}$ is the permittivity of the barrier). In the case when d=0 both charge distributions of electron and hole mutually cancel themselves resulting in neutral exciton (if one neglects here their recombination). The barrier prevent the recombination but causes that both charge densities do not perfectly cancel, and the resulting excitons interact with neighbors. This interaction is of repulsion type and reflects the residual repulsion of ordered states in stripes (Equation 10) both of electrons and of holes. Attraction may occur only on borders of stripes, which is negligibly small. This repulsion of indirect excitons created by Landau electrons and holes is very important—is needed for the superfluid transition of BEC [80].

Because of the degeneracy of the LLL, $\frac{BS^{\prime }e}{h}$ ($S^{\prime }$ is the surface of the stripe), the same in upper (electron) and bottom (hole) stripes all excitons have the same energy and the energy of binding of such excitons is also the same. Thus, these excitons form BEC and fall into superfluid state because of their repulsion, optimal in stripes. It can be handled upon the Bogolubov model of superfluidity developed to the binary fluid of excitons (cf. Appendix A), because in the case of twin Hall system with complementary filling, $\nu_{top}+\nu_{bot}=1$, we deal simultaneously with two polarizations of indirect excitons, ± and ∓, in neighboring alternately aligned horizontally stripes. Neglecting attraction near the borders of stripes, these two oppositely polarized exciton liquids are independent and mutually related only via complementary filling pattern. The two-component system differs from the single-component one [82,84,92], the latter being the simplified repetition of theoretical model of Landau particle excitons in a single layer (neglecting their instability) when due to d=0 polarization loses its sense [84].

Indirect excitons are the more stable the lower barrier thickness is (as their energy $∼-\frac{e^{2}}{d}$), but the thickness of the barrier, *d*, controls also the stability of the reentrant IQHE phase, thus the energy competition decides on the final phase diagram. It has been found according calculation scheme presented in Appendix A and the result is plotted in Figure 3b) [93]. In the experiments [19,20,56,73,76,78,79], the fully developed IQHE-reentrant behavior, with $R_{xx}=0$, and quantized, $R_{xy}=\frac{h}{e^{2}}$, is observed at total filling $\nu_{T}=\nu_{top}+\nu_{bot}=1$ (or higher integer) in the range of the entire diagonal of $\left(\nu_{top},\nu_{bot}\right)$. On the other hand, the presence of the superfluid coherent phase of BEC is demonstrated by the counterflow effect at lower *d* [19,20,56,73,74,75,76,77,78,79]. The phase diagram as in Figure 3b) is confirmed by the experiment. Worth noting the concave shape of phase transition curve between superfluid (lower) and IQHE-reentrant (upper) phases agrees quantitatively [93] with the experimental data both in GaAs [56] and double-bilayer graphene structure [19,20].

The superfluid state is observed in experiment also in twin Hall system configuration when both layers are locked-up on one end and the current driven in upper layer traverses through the bottom layer in the opposite direction [76]. Please note that holes in the LL stripes are in fact charges of net static jellium, thus negative stripes of electrons slip on the static jellium and the hole stripes do not contribute to the current and move in the same direction as emptying regions between moving electron stripes overfilled with electrons. The drive-current of electrons in upper layer causes the opposite directed current in reciprocal bottom layer at closed circuit. Such configuration, known as the counterflow one, dismisses, however, the stripe structure because stripes in both layers pass each other perturbing electrostatic energy trade-off considered for IQHE-reentrant state at the static situation. In the dynamical case of the counterflow configuration the IQHE phase is thus completely dismissed in favor of the superfluid state of excitons. The latter is robust against counterflow because striping for BEC of excitons is in the *k* space (*k* is the quantum number in Equation (Equation 10)) linked to the position space only up to the gauge-invariance (cf. Figure 5). This gauge-invariance causes that stripes for excitons have not determined orientation in the position space and thus are immune to the counterflow. Effectively, electrons moving in one direction in upper layer are still coupled with holes from the bottom layer, though electrons in the bottom layer move in the opposite direction. This contradicts the transverse structure of stripes but agrees with virtual striping in *k* space not oriented due to gauge-invariance for excitons. The counterflow configuration prefers thus the exciton BEC superfluidity and it is indeed observed experimentally in agreement with theoretical phase diagram presented in Figure 3c).

Another situation is in the so-called drag configuration when the layers are not connected at any point. The drive-current in the upper layer pulls electrons and their stripes in one direction in this layer. The electrically coupled alternatively aligned stripe structure in the bottom layer is dragged in the same direction allowing the same energy trade-off as in the static case. Both IQHE-reentrant state and exciton BEC superfluid are possible here and the phase diagram displays energy competition between these phases—as shown in the Figure 3b).

In the so-called drag configuration, which corresponds to the case when layers are not locked-up [76,94,95] (not closed circuit), the drive-current in one layer pulls a drag-voltage. In the counterflow configuration, when layer are locked-up, the superfluidity of the BEC of indirect excitons is believed to explain the observed drop to zero of $R_{xx}$ and $R_{xy}$ in both layers, whereas in the case of the drag (parallel) configuration $R_{xx}^{drive}=R_{xx}^{drag}=0$ and $R_{xy}^{drive}=R_{xy}^{drag}=\frac{h}{e^{2}}$ are referred rather to the IQH-reentrant correlation in the total system [19,20,56,75,76], cf. Figure 3.

The BEC of indirect excitons in a bilayer quantum Hall system with complementary filling, $\nu_{T}=\nu_{bot}+\nu_{top}=1$, has been researched theoretically in Ref. [82,92] using a former analysis of excitons in monolayer quantum Hall systems [81,82,83,84,92]. However, in contrast to the monolayer systems, in bilayer systems the indirect excitons acquire an additional degree of freedom in the form of the vertical polarization, $\left(\pm \right),\left(\mp \right)$, which was not define in the monolayer planar system. Therefore, in the bilayer system the superfluid BEC is thus a two-component liquid and not single-component as was in monolayer. Both these components are equally dynamical, which is not accounted properly for if one develops the direct analog to single-component BCS superconductor as in [92].

The creation of indirect excitons in the twin Hall system is related with the inter-layer Coulomb coupling across the barrier of Landau band electrons and holes in complementary states in reciprocal layers and numbered by the quantum numbers (k,n,λ) (written e.g., in the Landau gauge, A=(0,Bx,0)) as in Equation (Equation 10). Because the hole state (with α=−1 in Equation (Equation 10)) for −k has the same 2D charge distribution as the electron state (with α=1 in Equation (Equation 10)) with *k*, the e-h pairs, *k* and −k, are ideally neutral in the d=0 limit (*d* is the barrier thickness). the same pairs are assumed to create indirect excitons for d≠0 though in this case the separated charges do not compensate themselves mutually [82,84,92]. The residual charge of such excitons allows their interaction. Please note that the various internal structures of these excitons related to different *k* do not influence their energy due to the degeneracy of LLs which causes that all electrons and holes have the same energy. Therefore (k,−k) inter-layer excitons have all the same energy and can be considered to be the BEC. The excitons homogeneously polarized in the case of d≠0 repulse themselves, thus, they fall into the superfluid state according the BEC Bogolubov model [80]—cf. Appendix A where the two-component version of this model is presented.

The coupling between electrons k,1(2) and holes −k,2(1) from opposite layers leads to the BEC of these pairs at d≠0 in the case of the complementary filled layers, $\nu_{top}+\nu_{bot}=1$ (bot and top subscripts denote the pseudospin components λ=1,2), taking the advantage of the same spatial in-plane shape of states *k* (electron) and −k (hole) despite the different λ. To create the exciton, *k* electron and −k hole must be located one in front of another in the opposite layers. This requires, however, a specific spatial ordering of *k* electrons in one layer and the precisely oppositely ordered −k holes in the reciprocal layer. Such an ordering of *k* states which we will call as *k*-striping. S mentioned above, the indirect (k,−k) excitons interact in a residual manner due to the electrostatic scattering of the states (Equation 10) in each layer, because the charges of electrons and holes in the excitons are not ideally canceled due to the finite layer separation, d≠0 [82].

This residual interaction causes the repulsion of the excitons if they are uniformly polarized. When the *k* stripes correspond to the stripes with local filling $\nu_{local}=1$ alternately with $\nu_{local}=0$, complementary aligned in the top and bottom layers, then such a striping allows for pairing of all electrons holes and holes (electrons) in proper states in both layers and the simultaneous repulsing of excitons. Please note that the LL degeneracy in stripes with $\nu_{local}=1$ is lower than the LL degeneracy for the entire sample with a surface *S* due to the smaller surface $S^{\prime }$ of the stripe than *S*. The initial number of electrons for $\nu_{top(bot)}<1$ (referred to the LL degeneracy at the entire sample surface, $N_{0}=\frac{BSe}{h}$) is thus divided into subsets to fill stripes up to $\nu_{local}=1$ (corresponding to the local LL degeneracy in the stripe which equals to $N_{0}^{\prime }=\frac{BS^{\prime }e}{h}$). We see that the electrons (holes) in *k* stripes repulse themselves resulting in maximal repulsion of neighboring indirect excitons. This repulsion induces the superfluidity transition (according to the Bogolubov model) of both polarization components of excitons being already in BEC state. The related to BEC superfluid transition energy gain favors thus the *k* striping.

The indirect excitons consisting of electrons in states k,1(2) and a hole in states −k,2(1), are unique. They are not identical particles because of the internal structure (in general excitons are not true bosons in fact, and can be regarded as bosons only approximately). Nevertheless, all these indirect excitons have the same energy, thus can be considered to be two-component BEC [70,82,83,92]. In fact they are not true bosons as mentioned above and differ in their internal structure (k,−k) states by one per polarization. If to neglect the internal structure of theses excitons and to concentrate on their same energy they can be treated as the BEC.

From the geometry of the system it is clear that to maximize the exciton repulsion (convenient for the superfluid transition), the vertically complementary stripes must neighbor horizontally with the stripes with oppositely polarized excitons. The resulted whole the electrostatic structure is the subject for the energy minimization, including energy gain due to e-h coupling across the barrier and the competition with the reentrant IQHE correlation of carriers.

The repulsion of excitons causing the superfluidity of BEC [80,82,83] arises due to the spatial distribution of electrons and holes appropriately to both polarizations of excitons. The spatial distribution of Landau *k* states (at Landau gauge of magnetic field) (Equation 10) supports profiles stretched along the *y*-axis being of a Gaussian shape (modified, however, by the $w_{n}$ term for n>0)) in the transverse direction *x*, with perfectly ordered positions of the density maxima for succeeding *k* (Figure 4). In fact, the x,y orientation choice is arbitrary according to the freedom in selection of the magnetic field gauge (Figure 5). In order to assure the highest inter-layer electron-hole coupling and simultaneously the maximal repulsion of excitons, the complementary filled *k* states should be striped in the *k*-space perfectly inversely in both layers which corresponds to just complimentary striping with $\nu_{local}=1$ or 0 (this holds at any gauge of the magnetic field). Such striping in the *k*-space guarantees the formation of maximal repulsing excitons and the eventual optimal superfluidity of both oppositely polarized components of BEC, $\left(\pm \right)$ and $\left(\mp \right)$.

To demonstrate the above described scheme in more quantitative manner, we consider the stripe segment as marked in Figure 3a, where *L* is the transverse size of the layer and l=x+(l−x) is the length of the tested stripe segment. Here *x* means the length of empty stripe ($\nu_{local}=0$) and an (l−x) the length of the completely filled stripe ($\nu_{local}=1$) in the top layer and conversely in the bottom layer. This specific choice agrees with the initial nominal fillings, $\nu_{top}=\frac{l-x}{l}$, $\nu_{bot}=\frac{x}{l}$. Hence, the densities of charge in jellium in top and bottom layers are $\rho_{top}=\frac{l-x}{l2\pi }$, $\rho_{bot}=\frac{x}{l2\pi }$, correspondingly (they are expressed in units $\frac{\left|e\right|}{l_{B}^{2}}$ where *e* is the electron charge). The energy gain ΔE for to testing stripe sector for $\nu_{local}=1$ in filled stripes and thus with the reentrant IQHE correlation in these stripes with respect to the non-striped layers for this sector with nominal fractional fillings of both layers, $\nu_{top}=\frac{l-x}{l}$ and $\nu_{bot}=\frac{x}{l}$, has the form,
(11)$$\Delta E=\Delta E_{top}+\Delta E_{bot}+\Delta E_{inter}.$$
The terms in $\Delta E_{top(bot)}$ represent the electrostatic energy shift caused by striping internally in each layer, top and bottom, whereas $\Delta E_{inter}$ account for the contribution to the energy gain due to the inter-layer interaction, which, including the energy gain due to IQHE correlations in the completely filled $\nu_{local}=1$ stripes, stabilizes the reentrant IQHE. The explicit forms of these terms are listed in Appendix A. With the reentrant IQHE competes the superfluid BEC of excitons described above. Indirect excitons also take an advantage from the complementary striping structure in both layers but with energy gain due to coupling of electron-hole pairs and energy due to the superfluid transition (cf. Appendix A).

One can notice that the condition $\nu_{T}=1$ (or another integer) leads to the unique situation when two stripes in the top layer, the first one of width *x* and empty of electrons (and thus positively charged due to jellium) and the subsequent one of (l−x) width overfilled with electrons up to local $\nu_{local}=1$, ideally fit the inverted stripe ordering in the opposite bottom layer, as shown in Figure 3a). The negatively charged stripes are overfilled with electrons (beyond the jellium charge still corresponding to the nominal $\nu_{top}=\frac{l-x}{l}$ and $\nu_{bot}=\frac{x}{l}$) up to the local $\nu_{local}=1$ (with the density of electrons $\frac{1}{2\pi }$ per $l_{B}^{2}$ surface; the jellium charge density equals to $\frac{\nu_{nom}}{2\pi }$).

The numerical analysis of Equation (Equation 11) (for particularities cf. Appendix A) using the Metropolis Monte Carlo assessment of IQHE-reentrant energy [6,7] gain is sketched in Figure 3b,c, where the phase diagrams for the GaAs/GaAlAs/GaAs are shown. Similar diagrams o can be found for bilayer graphene/hBN/bilayer graphene and are presented in more detail in [93]. The energy gain caused by the coupling of excitons equals to by $\Delta E_{1}≃\frac{L}{2\pi \varepsilon_{2}d}$ in units $\frac{e^{2}}{4\pi \varepsilon_{0}l_{B}}$ (it is overestimated a bit as taken for point charges, but this fault is compensated by the energy gain due to superfluid transition). To the energy of the BEC phase contributes also the spatial redistribution of electrons and holes in the complementary manner in both layers in order to optimize the repulsion of indirect excitons in *k* stripes, which is required to superfluid transition in BEC.

The dependence of the energy gain with respect to *d* and *x* for both competing phases is shown in Figure 3b). One can observe that the reentrant IQHE state is energetically stable for any values of 0<x<l (i.e., for nominal fillings $\nu_{top}=\left(l-x\right)/l$, $\nu_{bot}=x/l$), provided that $d^{**}<d<d^{*}$, $d^{*}≃8l_{B}$ (GaAs and GaAlAs barrier) $4l_{B}$ (bilayer graphene and hBN barrier) [93], while at $d=d^{**}≃1.65-2.2l_{B}$ (GaAs) $0.8-1.2l_{B}$ (bilayer graphene) (depending on *x* and on l/L) the transition to the superfluid BEC phase takes place. For GaAs/GaAlAs/GaAs we assumed $l=100l_{B}$ and $L=25l_{B}$ in order to adjust $d^{**}$ to the experimentally observed value, $d^{**}=1.65l_{B}$ at the balanced filling as has been reported in Ref. [56]. The concavity of the phase transition curve for transition of the IQHE-reentrant state to the BEC of excitons, as plotted in Figure 3b) well agrees with the experimental data [19,20,56], in which the activation energy of the BEC phase exhibits an increase with the deviation away from the balanced filling and at diminishing *d* as well.

One ca observe also that for $d>d^{*}$ (at balanced filling), the IQHE-reentrant phase does not reach the symmetric filling, i.e., the balanced symmetrical filling $\nu_{top}=\nu_{bot}=\frac{1}{2}$ cannot be attained, however, the IQHE-reentrant state is still energetically lower for the imbalanced nonsymmetrical fillings of both layers, as is visible in Figure 3b).

According to the discussion presented above, the counterflow configuration the FQHE reentrant state is completely dismissed by the violation of the stripe structure in this configuration. Nevertheless, the *k*-space striping of excitons is not affected by the charge kinetics, as the *k*-stripe orientation is not actually fixed due to the gauge-invariance, thus, this orientation can be, without any loss of generality, chosen e.g., as directed parallel with the flow. Therefore, in the counterflow configuration, the BEC superfluid phase can be stable even for $d>d^{**}$ up to some value $d^{***}$ i.e., when the hyperplane of energy gain for superfluid exciton BEC crosses the zeroth energy plane, as is illustrated in Figure 3c).

Satisfactory consistence of the phase diagram in Figure 3 with experimental observations [19,20,56] demonstrates correctness of the model. Worth noting is the quantitative agreement of the activation energy for superfluid phase measured in double GaAs system with our prediction, being by one order lower than former estimation of this energy [56]. The latter was overestimated because assumed as the energy gain of superfluid phase with respect to the uncorrelated state, whereas the realistic value of the activation energy corresponds to the energy gain of superfluid phase with respect to the IQHE-reentrant state, formerly not accounted for.

The presented model explained also a puzzling observation [19] that in the twin bilayer graphene Hall system the exciton BEC superfluid is observed in the case when are complementary and fractionally filled LLLs of two bilayer graphene Hall subsystems, but only with Landau index n=0 and not n=1. In bilayer graphene 8-fold degeneracy of the LLL is related to 4-fold spin-valley degeneracy and so-called accidental degeneracy of Landau states with indices n=0 and n=1 [70]. Depending on the way of degeneracy lifting, the sub-bands with n=0 or n=1 can take part in the complementary filling of top and bottom Hall subsystems consisting of bilayer graphene sheets, $\nu_{top}+\nu_{bot}=1$. Disappearance of the superfluid signature in the case, n=1, in contrast to the full counterflow development at n=0, is very astonishing [19]. We have found explanation of this phenomenon upon the presented Bogolubov two-component model for indirect exciton BEC. For n=1 the states given by Equation (Equation 10) differ in the Hermite polynomial from those at n=0. Profiles of the charge density for the states (Equation 10) and for n=0,1,2 are depicted in Figure 6. The absence of the central peak for n=1 (odd) in contrast to n=0 or n=2 (even) causes that the repulsion of neighboring Landau particles in *k* stripes is ca. 4-fold lower for n=1 than for n=0 (or n=2). This repulsion is calculated as the integral over charge densities of neighboring particles and the overlap of these densities for n=1 strongly reduces the repulsion due to the absence of central peak of $H_{1}$ polynomial in contrast to $H_{0}$ or $H_{2}$. The resulting smaller repulsion of neighboring excitons destabilizes the superfluid phase (as visible in the Bogolubov model—cf. Appendix A). Disappearance of the superfluidity at n=1 emphasizes the crucial role of the exciton repulsion. Its lowering by the factor 4 dismisses the superfluid phase. To assure sufficient strong repulsion of excitons the *k* striping is thus necessary. Diluted (non-striped) distribution of excitons at complementary fractional filling of top and bottom layers strongly diminishes the repulsion of excitons for both their polarizations. Only the maximal value of this repulsion warranted at *k* striping of electrons and holes admits the superfluid transition.

## 4. Conclusions

In twin Hall systems various types of Hall correlation organization occur depending on the regime of coupling between layers and on the inter-layer tunneling of electrons. These regimes are defined by the inter-layer tunneling of electrons, which can be admitted or blocked. In the case when this inter-layer tunneling of electrons is admitted, the homotopy phases of the bilayer system are essentially different in comparison to the monolayer case. The braid homotopy patterns and related correlations reflect hopping of braid trajectory among layers which modifies the commensurability conditions and resulted correlations. Such a behavior is observed in bilayer graphene and in GaAs double systems when the barrier between layers is thin and low. These correlations of topological character and conditioned by the tunneling of electrons between layers causing also the hopping of cyclotron loops between layer, disappear, however, if the electron tunneling is blocked. The blockade of the braid trajectory hopping may be imposed in the bilayer system by various ways, e.g., by the widening of the barrier separating layers or by the application of a vertical voltage between layers or even by the tilting of the magnetic field. The influence of such factors on the homotopy of trajectories was experimentally demonstrated.

In the case when electron tunneling is blocked, a Coulomb coupling of carries across the barrier can be still possible for barrier thickness of order of the magnetic field length. Though the topology of the Hall system is of monolayer type, the Coulomb coupling may induce various specific correlations of electrons at a special complementary filling of the system, $\nu_{T}=\nu_{bot}+\nu_{top}=1$ (or other integer). In this case the alternately striping of both layers is shown to be energetically convenient because of the inter-layer Coulomb attraction of oppositely charged stripes. This energy gain allows for stability of IQHE-reentrant in locally filled stripes with $\nu_{local}=1$ despite arbitrary nominal filling of layer, provided that the complementary condition is fulfilled. Such a Hall correlation arrangement is experimentally observed in GaAs/GaAlAs/GaAs for the insulating GaAlAs barrier thickness exceeding ca. $1.6l_{B}$ and in b-graphene/hBN/b-graphene for hBN layer thickness exceeding $l_{B}$, in the so-called drag configuration of a bilayer setup. This IQHE-reentrant correlated state competes, however, with the different collective effect, namely with the coherent state of inter-layer indirect excitons created by binding of Landau electrons and holes from opposite layers. If electrons and holes from opposite layers occupy the mirrored Landau states and are located exactly one opposite the other, the indirect excitons can be formed all with the same energy due to the degeneracy of LLs. This coherent state occurs to be superfluid which was demonstrated experimentally by the counterflow observation in the so-called counterflow configuration of a twin Hall systems complementary filled with electrons. This superfluid phase of excitons occurs to be more convenient energetically at lower thickness of the insulator barrier, however thick enough to block the inter-layer tunneling of electrons and the recombination of electrons. This superfluid state was experimentally observed in b-graphene/hBN/b–graphene and in GaAs/GaAlAs/GaAs twin Hall systems. The observations are consistent with the theoretical phase diagram for competition of the superfluid state and the IQHE-reentrant state in the same bilayer system.

The theoretical phase diagram agrees with the experimental observations for the twin GaAs and graphene heterostructures. In the drag configuration, both phases occur: for $d<d^{**}$ (*d*–the barrier thickness) the indirect exciton superfluid phase is more stable, while for $d^{**}<d<d^{*}$ the reentrant IQHE state energetically prevails. The values of the critical barrier thickness depend on the barrier and layer material properties (especially on their dielectric permittivities) which agrees with the experimental data for GaAs/GaAlAs/GaAs and bilayer graphene/hBN/bilayer graphene. The energy trade-off due to the intra and inter-layer Coulomb interaction in the static configuration breaks down, however, in the case of the dynamic counterflow configuration, which destabilizes the reentrant IQHE phase in favor of the exciton superfluid phase. This also agrees with the experimental observations.

Both regimes for electron tunneling between layers in stacked Hall systems offer thus possibility for control over topological homotopy phases defining correlation patterns, which would be convenient for QIP on the platform of multilayer stacked Hall systems. The great advantage of such a platform is the related already highly advanced sampling techniques and very well developed multi-electrode measurement methods—all described above correlation homotopy properties have been already demonstrated experimentally both in GaAS multi-layers as well as in graphene realizations.

## Figures and Tables

**Figure 1 nanomaterials-10-01286-f001:**
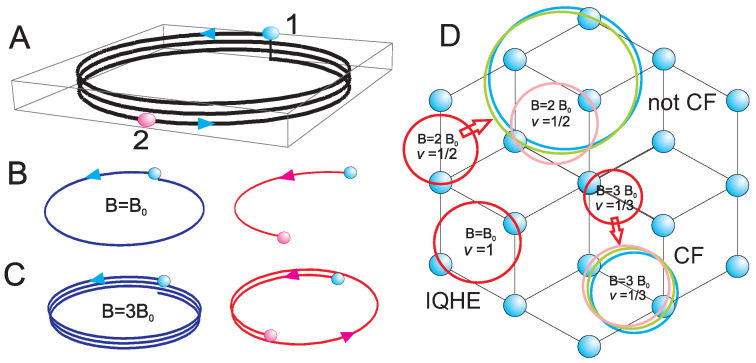
(**A**) An illustration of the commensurability of the three-loop cyclotron orbit with the separation of two closest particles, shown as blue and red. (**B**) An illustration of single-loop cyclotron orbit (left) and related braid for exchange of two closest particles, shown as blue one red—the braid is built from the half of the cyclotron orbit. (**C**) An illustration of thee-loop cyclotron orbit (left) and related braid for the exchange of two particles blue and red—in this case, the braid gains one additional loop (the third dimension added for clarity of the vision, more detail in Ref. [5]). (**D**) An illustration of the commensurability in Wigner-type lattice of electrons in graphene monolayer. Several situations are illustrated. The single-loop cyclotron orbit commensurability with nearest neighbors corresponding to IQHE at field $B_{0}$. At three-times larger field $3B_{0}$ the single-loop orbit is too short to match nearest neighbors in the Wigner lattice—but the three-loop orbit fits to nearest neighbors. This corresponds to FQHE at $\nu =\frac{1}{3}$. For field $2B_{0}$ single-loop orbit is also too short in comparison to the electron separation. In this case the homotopy pattern with $x_{1}=1,\;x_{2}=2,\;x_{3}=2$ (cf. Table 2) allows fulfillment of the commensurability condition but for next-nearest neighbors for two loops—hence, this state is of FQHE type at $\nu =\frac{1}{2}$, but not accessible for the CF model. The envelope function of multiparticle states in graphene is distinct than the envelope function in GaAs. In graphene this envelope function favors next-nearest neighbor correlations, which manifest itself in stability in graphene of FQHE at $\nu =\frac{1}{2}$ beyond the CF model [47,55].

**Figure 2 nanomaterials-10-01286-f002:**
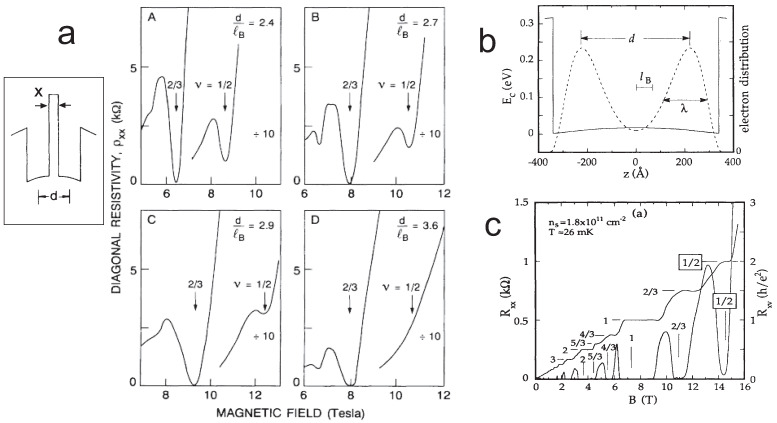
Astonishing observations of FQHE at 1/2 in bilayer GaAs 2DES. (**a**) Observation of ν=1/2 FQHE state in double well GaAs/GaAlAs/GaAs (left inset) which is dependent on the width of the barrier *x* as listed in Table 3, after [45]. (**c**) The similar observation of FQHE at 1/2 in bilayer Hall structure in wider well in GaAs heterostructure with transverse profile of the electron density shown in (**b**) (the profile of the well with width 68 nm used in [44] to separate electron Hall layers—solid line indicates the well conduction band, dashed line shows electron distribution for $n=1.8\times 10^{11}$/cm${}^{2}$, d=44.5 nm, the magnetic length $l_{B}$ is marked for B=15 T at which the state $\nu =\frac{1}{2}$ is observed); the state at 1/2 disappears when the well is too wide resulting in the inter-layer barrier growth, after [44].

**Figure 3 nanomaterials-10-01286-f003:**
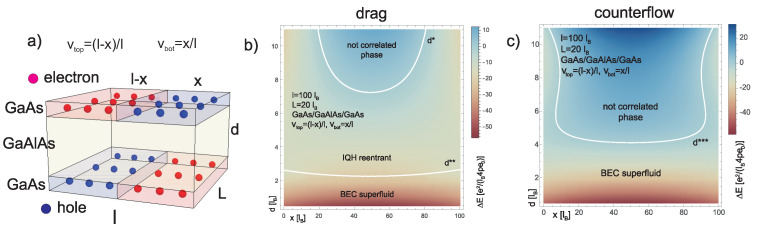
(**a**) The striped sector in the bilayer Hall structure used for trade-off energy of striping cost against the energy gain due to IQHE-reentrant correlation or superfluid exciton BEC state, with marked stripe dimensions. Electrons are shifted to the over-charged stripes, while empty stripes are hole stripes charged by the positive jellium. The structure is mirrored symmetrical with charge change across the barrier of thickness *d*, which is possible only for a complementary filling of layers, $\nu_{top}+\nu_{bot}=1$ (r other integer), with $\nu_{top}=\frac{x}{l}$ and $\nu_{bot}=\frac{l-x}{l}$ (in the figure, both equal to 1/2). (**b**) The resulted phase diagram, versus barrier thickness *d* and the imbalanced complementary filling pattern (cf. Appendix A) with $\nu_{top}=\frac{x}{l}$, for the static or drag configuration found by the MMC simulation in GaAs/GaAlAs/GaAs bilayer structure. (**c**) The phase diagram for the same system at the counterflow configuration, when the IQHE-reentrant state is completely dismissed.

**Figure 4 nanomaterials-10-01286-f004:**
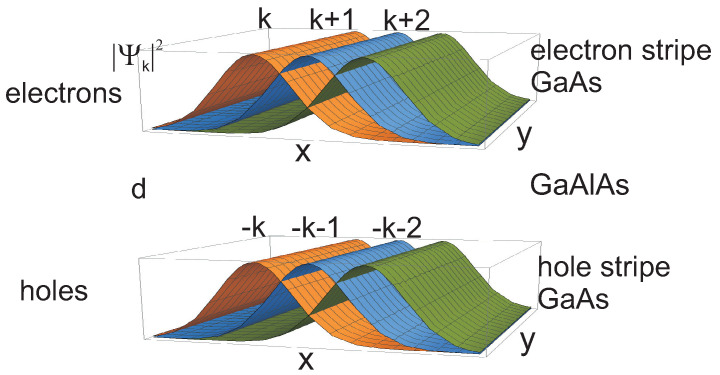
Pictorial presentation of *k* striping of electrons and holes in opposite layers. Charge density of the electron, according to wave functions given by Equation (Equation 10) for *k* state, is exactly the same as of the hole in −k state. To allow exciton formation, the consecutive states k,k+1,k+2,… ordered as represented by their density, $|\Psi_{k}{|}^{2}$, in the figure in compliance with Equation (Equation 10) must by aligned perfectly in front of oppositely ordered states of holes, −k,−k−1,−k−2,….

**Figure 5 nanomaterials-10-01286-f005:**
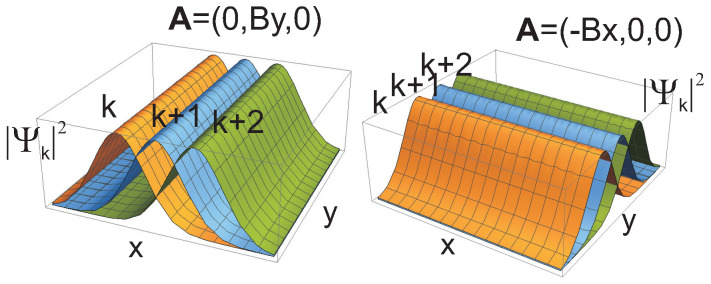
Depending on the gauge of magnetic field *B* the shape of the electron and hole densities, as given by Equation (Equation 10) for the Landau gauge, changes as shown in the figure for two variants of the gauge, A=(0,Bx,0) and A=(−By,0,0). The striping in the *k* space is not attributed thus to fixed orientation, due to the gauge invariance.

**Figure 6 nanomaterials-10-01286-f006:**
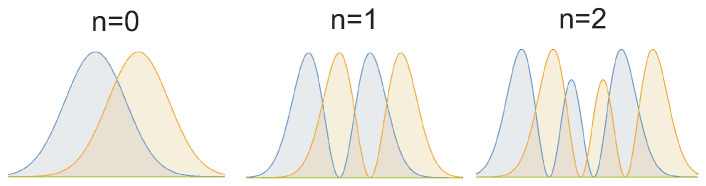
(left) Profiles of the densities of states given by Equation (Equation 10) for neighboring electrons with *k* and k+1 for n=0,1,2, where *n* is the Landau index enumerating the Hermite polynomial in Equation (Equation 10). For *n* odd the central peak disappears, which diminishes by factor ca. 4 the repulsion of two successive electrons (holes) in the *k* stripes.

**Table 1 nanomaterials-10-01286-t001:** Classification of multi-electron homotopy phases [m(b)GN—monolayer(bilayer)-graphene, ChTI—Chern topological insulator].

Dim.	System	Braid Metrics and Type of Nesting	Homotopy Phases
3D	gas or interacting electrons	none braid metrics	none homotopy correlation phases except for Pauli correlations
2D	gas	none cyclotron braid nesting	none homotopy correlation phases except for Pauli correlations
2D	interacting electrons	single-loop cyclotron braid nesting	homotopy phases of IQHE, $\nu =\frac{N}{N_{0}}=1,2,\ldots$; FQHE in higher LLs, ν=2+(1[i])+1/3(2/3),4[6]+(1[i]) + 1/5(…, 4/5) (GaAs), [mGN, i=1,2,3] [6,7,53]
2D	interacting electrons	multi-loop cyclotron braid nesting	homotopy phases of FQHE, $\nu =\frac{xy}{(q-1)y\pm x}$ [6]
bGN 2D-2D	interacting electrons	multi-loop cyclotron braid nesting	homotopy phases of FQHE, inter-layer distribution of loops and inter-layer flux leakage [7]
ChTI 2D	gas	none Berry braid [4] nesting	none homotopy correlation phases except for Pauli correlations
ChTI 2D	interacting electrons	single-loop Berry braid nesting	homotopy phases of IChTI (integer ChTI), $\nu =\frac{N}{n_{0}}=1$, $n_{0}$ number of nodes
ChTI 2D	interacting electrons	multi-loop Berry braid nesting	homotopy phases of FTChI (fractional ChTI), $\nu =\frac{xy}{(q-1)y\pm x}$

**Table 2 nanomaterials-10-01286-t002:** Some selected homotopy patterns according to Equation (Equation 6) and the corresponding filling rates ν, $x_{i}$ calculated according Equation (Equation 6).

$x_{1},\;x_{2},\;x_{3}$	$\nu =N/N_{0}$	type
1, 1, 1	1/3	CF
1, 1, 2	2/5	CF
1, 2, 2	1/2	not CF
$x_{1},\;x_{2},\;x_{3},\;x_{4},\;x_{5}$	$\nu =N/N_{0}$	type
1, 2, 2, 2, 2	1/3	not CF
1, 2, 2, 2, −2	1/2	not CF
1, 1, 1, 2, 2	1/4	not CF
1, 1, 1, 1, −1	1/3	CF
1, 1, 1, 2, −2	1/3	not CF
1, 1, 2, 2, −2	2/5	not CF
$x_{1},\;x_{2},\;x_{3},\;x_{4},\;x_{5},\;x_{6},\;x_{7}$	$\nu =N/N_{0}$	type
1, 2, 2, 2, 2, 2, 2	1/4	not CF

**Table 3 nanomaterials-10-01286-t003:** The collection of the parameters for the double well shown in the inset in Figure 2, *x* is the barrier thickness, *d* is the well center-to-center distance, for A−C*d* equals to 21.1 nm and for *D* to 27.9 nm, $l_{B}=\sqrt{\frac{\hbar }{eB}}$, both quantum well widths are 18 nm (after [45]).

No.	*x* (nm)	*n* ($10^{11}$/cm${}^{2}$)	$d/l_{B}$ (ν=1/2)	Strength (ν=1/2)
A	3.1	1.04	2.4	strongest
B	3.1	1.29	2.7	strong
C	3.1	1.52	2.9	weak
D	9.9	1.31	3.6	absent

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
