# Peer review of "Homotopy Phases of FQHE with Long-Range Quantum Entanglement in Monolayer and Bilayer Hall Systems"

_nanomaterials, 2020, doi:10.3390/nano10071286_

Round 1

Reviewer 1 Report

The author presents some interesting insights on the nature of quantum Hall phases in monolayer and multi-layer Hall systems. It is known that quantum Hall effect phases have a topological nature and they represent incompressible quantum fluids of their own kind. As such, these phases have been studied at various regimes and filling factors. Multilayer configurations create the opportunity to change these topological phases in a tunable way by introducing a new variable into the problem (inter-layer coupling). Many times, phases that are not observed in monolayer phases can be seen in multi-layer (bi-layer) systems. This manuscript represents a valid effort to categorize the various possibilities that may arise in this regime. Overall, the work is interesting and I would recommend publication after the author addresses the following points:

A) Page 2, line 41-42: The author has written that “Integer and fractional quantum Hall effects (IQHE/FQHE) are not assigned by any local order parameter and any broken symmetry.”

I strongly recommend adding the following sentence immediately after the prior statement:

“Exceptions are various model anisotropic quantum Hall states that break the rotational symmetry of the system [1-5].”

[1] Phys. Rev. B 65, 045306 (2001).

[2] Phys. Rev. B 67, 155315 (2003).

[3] Phys. Rev. B 69, 125320 (2004).

[4] Phys. Rev. B 85, 115308 (2012).

[5] Phys. Rev. B 85, 165318 (2012).

B) There are few typos that should be corrected either by the author, or during the proof-reading process. I mention a pair of them: page 1, line 30: correct from “attemts” to “attempts”; page 1, line 32: correct from “Barry” to “Berry” and few others like these.

The manuscript will be improved by the addition of discussions and references mentioned above. I would recommend publication of the manuscript after the authors amend the manuscript to add discussions and references that address the points listed above.

Author Response

Thank you for a positive opinion. The suggested corrections have been done.

A. The phrase, "Exceptions are various model anisotropic quantum Hall states that break the rotational symmetry of the system [1-5].” has been added in line 42 (page 2)

B. Several typos are corrected, including those indicated by the Referee

C. The suggested references have been added with the comment as suggested in A.

All recommendations are fulfilled and corrections are shown in red color in the resubmitted PDF file.  

Reviewer 2 Report

In the present paper the author considers multilayer Hall systems which can be realized both in GaAs heterostructures and in graphene.By properly tuning the coupling between the layers and the interaction within the layers new and interesting phases can be achieved.

Considering a topological homotopy approach author is able to go beyond the conventional composite fermion description of fractional quantum Hall systems. 

In my opinion the paper is very interesting, scientifically sound, well written and, even if quite long, readable and well organized. It open new perspective in this field of research. 

Therefore, I think it deserves to be published on Nanomaterials in the present form. 

I only would like to suggest the author to have a carefully spell-check of the manuscript. I have identified various typos that should be avoided. Only to name few: 

- Page 1: “attemts”

- Page 3: “quuantum”, “bilayr”

- Page 4: “commnsurability”

- Page 11: “fist”

- Page 16: “trough”

- Throughout the whole paper the notation multiloop/molti-loop (and similar) are used. I think it is better to consistently choose one of them. 

Author Response

Thank you very much for a very nice opinion.

The suggested spell-check has been performed.

The indicated typos (and several others) have been corrected and corrections are indicated by the red color throughout the text.

The term singleloop (and similar) is substituted by single-loop.  

Reviewer 3 Report

The authors have studied the topological phases state and the possibility to control them in multilayer Hall systems. Previous experimental results are analyzed

The subject is interesting however before giving my final decision, some critical remarks should be answered and taken into consideration for the revised version:

  • The authors introduce several concepts briefly such as quantum entanglement. The authors should write an extended paragraph on the entanglement and it would be even better about quantum correlations in physical systems since the quantum entanglement is a special case of quantum correlations, this will increase the visibility of the paper. The following references are helpful: PRA 91, 032309 (2015); Quant. Inf.   12, 69 (2013); PRA 84, 053817 (2011), and references therein.  
  • The authors should clarify more in details the derivation of eq 6, they only mentioned the reference “Due to energy preference discussed in [6]”
  • Some misprints should be cleaned such as “quuantum entangled” it should be corrected “quantum entanglement” or just “entanglement”
  • The authors mentioned several times the FQHE states, it should be a detailed discussion to introduce it.  

Author Response

Thank you for the positive opinion and the suggestions how to improve the presentation.

According to these suggestions we have added the paragraph about quantum entanglement and correlations in the page 3 with the suggested references included. This paragraph is shown in the red color in the resubmitted PDF file.

According to Eq.(6) the short explanation is added, just near this equation (as shown in the red color in the PDF file).

Several missprints and errors  have been removed (including those indicated by the Referee).

The term 'FQHE state' has been introduced and shortly commented (as shown in the red color in the revised version of PDF file in the page 2).

Round 2

Reviewer 3 Report

The revised version is ok.